# The Overlapping Biology of Sepsis and Cancer and Therapeutic Implications

**DOI:** 10.3390/biomedicines13061280

**Published:** 2025-05-23

**Authors:** Amit Kumar Tripathi, Yogesh Srivastava

**Affiliations:** 1Department of Microbiology, Immunology and Genetics, University of North Texas Health Science Center, Fort Worth, TX 76107, USA; 2Department of Genetics, University of Texas MD Anderson Cancer Center, Houston, TX 77030, USA

**Keywords:** sepsis, cancer, inflammation, immunosuppression, tumor microenvironment, cytokines, immune response, therapeutic targets

## Abstract

Sepsis and cancer, though distinct in their clinical manifestations, share profound pathophysiological overlaps that underscore their interconnectedness in disease progression and outcomes. Here we discuss the intricate biological mechanisms linking these two conditions, focusing on the roles of inflammation, immune dysregulation, and metabolic alterations. In sepsis, an uncontrolled immune response to infection leads to a cytokine storm, tissue damage, and immune paralysis, while cancer exploits chronic inflammation and immunosuppressive pathways to promote tumor growth and metastasis. Both conditions exhibit metabolic reprogramming, such as the Warburg effect in cancer and glycolysis-driven immune cell activation in sepsis, which fuels disease progression and complicates treatment. Sepsis can exacerbate cancer progression by inducing genomic instability, epigenetic modifications, and a pro-tumorigenic microenvironment, while cancer increases susceptibility to sepsis through immunosuppression and treatment-related complications. The shared pathways between sepsis and cancer present unique opportunities for therapeutic intervention, including anti-inflammatory agents, immune checkpoint inhibitors, and metabolic modulators. Anti-inflammatory therapies, such as IL-6 and TNF-α inhibitors, show promise in mitigating inflammation, while immune checkpoint inhibitors like anti-PD-1 and anti-CTLA-4 antibodies are being explored to restore immune function in sepsis and enhance antitumor immunity in cancer. Metabolic modulators, including glycolysis and glutaminolysis inhibitors, target the metabolic reprogramming common to both conditions, though their dual roles in normal and pathological processes necessitate careful consideration. Additionally, antimicrobial peptides (AMPs) represent a versatile therapeutic option with their dual antimicrobial and antitumor properties. In this review, we also highlight the critical need for integrated approaches to understanding and managing the complex interactions between sepsis and cancer. By bridging the gap between sepsis and cancer research, this work aims to inspire interdisciplinary collaboration and advance the development of targeted therapies that address the shared mechanisms driving these devastating diseases. Ultimately, these insights may pave the way for novel diagnostic tools and therapeutic strategies to improve outcomes for patients affected by both conditions.

## 1. Introduction

Sepsis and cancer represent two major global health burdens with complex interconnections. Sepsis affects nearly 50 million people annually, causing over 11 million deaths worldwide, while cancer accounts for 9.7 million annual deaths, with projections suggesting a 77% increase in cases by 2050 [1]. Despite advances in antimicrobial therapies and critical care, sepsis remains a formidable clinical challenge due to its complex pathophysiology and high risk of multi-organ failure. On the other hand, cancer, characterized by uncontrolled cell proliferation and metastasis, is a leading cause of death globally, with an estimated 10 million deaths in 2020 alone [2]. The burden of cancer continues to rise, driven by aging populations, lifestyle factors, and environmental exposures. While sepsis and cancer are traditionally viewed as distinct entities, emerging evidence suggests a profound and bidirectional relationship between the two conditions [3,4]. Sepsis can act as a trigger for cancer progression by inducing chronic inflammation, immune suppression, and genomic instability [4,5]. Conversely, cancer patients are at heightened risk of developing sepsis due to immunosuppression caused by the malignancy itself or its treatments, such as chemotherapy, radiotherapy, and surgery [6]. This interplay between sepsis and cancer has significant implications for patient outcomes, complicating diagnosis, treatment, and long-term prognosis [7]. The growing recognition of this bidirectional relationship has sparked interest in understanding the shared mechanisms underlying both conditions. Chronic inflammation, immune dysregulation, and metabolic alterations are common features of sepsis and cancer, suggesting potential overlapping therapeutic targets [8]. Moreover, the clinical management of patients with both sepsis and cancer presents unique challenges, necessitating a nuanced approach to balance immune activation and suppression. By synthesizing current evidence and identifying gaps in knowledge, we aim to provide a thorough understanding of the sepsis–cancer connection. Ultimately, this knowledge may pave the way for novel diagnostic tools and therapeutic strategies to improve outcomes for patients affected by these intertwined conditions.

## 2. Pathophysiological Overlap Between Sepsis and Cancer and Therapeutic Implications

Both sepsis and cancer share complex biological processes involving inflammation, immune system dysfunction, and metabolic alterations. Understanding these overlapping mechanisms not only elucidates their interconnectedness but also opens new possibilities for treating both conditions. Pathologically, Extracellular Toll-like receptors (TLRs), including TLR1, TLR2, and TLR4, are located on the cell surface and recognize bacterial pathogens, initiating signaling pathways that lead to dysregulated production of cytokines, chemokines, and interferons (IFNs). This excessive immune activation can result in a cytokine storm, which is a central driver of sepsis. Endosomal TLRs—such as TLR3, TLR7, TLR8, and TLR9—are situated within endosomes and detect bacterial and viral genetic material, further amplifying immune responses. Endogenous negative regulators of TLR signaling act to suppress overactivation of these pathways. However, when this regulatory control fails, unchecked TLR signaling leads to systemic inflammation, sepsis, and ultimately, death (Figure 1).

### 2.1. Cytokine Storm in Sepsis

In sepsis, the immune system’s response to infection can become dysregulated, leading to a phenomenon known as the “cytokine storm” [9]. Chronic inflammation in sepsis is also associated with the development of long-term complications, such as post-sepsis syndrome, which includes cognitive impairment, physical disability, and an increased risk of late mortality [10]. The hyperinflammatory phase in sepsis is characterized by the excessive and uncontrolled release of pro-inflammatory cytokines, such as interleukin-6 (IL-6) and tumor necrosis factor-alpha (TNF-α). These cytokines, which are essential for coordinating the immune response, can spiral out of control, causing widespread systemic inflammation, tissue damage, and multi-organ failure [11,12]. For example, IL-6 plays a critical role in the acute phase response, but its overproduction during sepsis can lead to endothelial dysfunction, vascular leakage, and hypotension, contributing to septic shock [13]. Similarly, TNF-α, a potent pro-inflammatory cytokine, can induce apoptosis in endothelial cells and disrupt the vascular barrier, exacerbating organ damage [14]. Studies have shown that elevated levels of IL-6 and TNF-α are strongly associated with poor outcomes in sepsis patients, highlighting their central role in the pathophysiology of the disease [10,15].

### 2.2. Cytokines in Cancer

In cancer, the same cytokines that drive inflammation in sepsis are often co-opted by tumors to promote their growth and survival. For instance, cancer cells can produce cytokines and chemokines that recruit immune cells, such as macrophages and neutrophils, to the tumor site. These immune cells, in turn, release additional cytokines and growth factors that promote tumor cell proliferation, angiogenesis, and invasion [16]. The inflammatory microenvironment also plays a key role in immune evasion, a hallmark of cancer. Tumors can manipulate the immune system to suppress antitumor responses and create an immunosuppressive milieu. It has been observed that tumor-associated macrophages (TAMs) and myeloid-derived suppressor cells (MDSCs) are often recruited to the tumor site, where they release immunosuppressive cytokines, such as IL-10 and transforming growth factor-beta (TGF-β), which inhibit the activity of cytotoxic T cells and natural killer (NK) cells [17]. Other cytokines such as IL-6 in cancer could be a potent growth factor for many types of cancer cells [18]. They activate signaling pathways such as JAK/STAT, which promote cell proliferation, inhibit apoptosis, and enhance angiogenesis. In ovarian cancer, elevated IL-6 levels have been linked to tumor progression and poor prognosis [19]. TNF-α, while initially identified for its ability to induce tumor cell death, has a more complex role in cancer biology. In some contexts, TNF-α promotes chronic inflammation, which creates a tumor-friendly microenvironment. This chronic inflammation may lead to DNA damage, genomic instability, and the activation of oncogenic pathways, all of which contribute to cancer initiation and progression [20]. Specifically, in colorectal cancer, TNF-α has been shown to enhance the invasive and metastatic potential of cancer cells by promoting epithelial–mesenchymal transition (EMT) [21]. Thus, both cancer and sepsis can lead to chronic inflammation which is a persistent, low-grade inflammatory state that is a common feature of both sepsis and cancer (Figure 2).

The central role of cytokines like IL-6 and TNF-α in both sepsis and cancer has led to the development of targeted therapies aimed at modulating their activity. For example, monoclonal antibodies against IL-6 or its receptor, such as tocilizumab, have shown efficacy in dampening the cytokine storm in sepsis and are also being explored as treatments for certain cancers [22]. In sepsis, tocilizumab has been investigated in clinical trials for its potential to reduce inflammation and improve outcomes [23]. Similarly, TNF-α inhibitors, which are widely used in autoimmune diseases, are being investigated for their potential to reduce inflammation-driven tumor progression in cancer [24].

However, the dual roles of these cytokines in both protective and pathological processes pose significant challenges. For instance, while inhibiting IL-6 or TNF-α may reduce inflammation and tumor growth, it could also compromise the body’s ability to fight infections or control tumor progression [25]. This highlights the need for a nuanced approach to therapy, one that balances the need to control inflammation without compromising essential immune functions (Table 1).

## 3. Immunosuppression in Sepsis and Cancer and Therapeutic Implications

### 3.1. Immune Paralysis in Sepsis

Sepsis typically induces a biphasic immune response: an initial hyperinflammatory phase followed by a prolonged state of immune suppression, known as “immune paralysis”. During this phase, the immune system becomes less effective at fighting infections, and patients are highly susceptible to secondary infections [26]. Immune paralysis is characterized by the dysfunction of key immune cells, such as T-cells, B-cells, and dendritic cells, which are essential for mounting an effective immune response [27].

T-cells, in particular, play a critical role in the adaptive immune response, but in sepsis, they often become anergic or exhausted, losing their ability to proliferate and produce cytokines [28]. This T cell dysfunction is partly mediated by the upregulation of immune checkpoint molecules, such as programmed cell death protein 1 (PD-1) and cytotoxic T-lymphocyte-associated protein 4 (CTLA-4), which act as brakes on the immune system. For example, PD-1 is upregulated on T cells during sepsis, and its interaction with its ligand PD-L1 inhibits T cell activation and promotes T cell exhaustion [29]. Similarly, CTLA-4, which is expressed on regulatory T cells (Tregs) and activated T cells, can suppress T cell responses by competing with the co-stimulatory molecule CD28 for binding to B7 molecules on antigen-presenting cells [30].

In addition to T-cell dysfunction, sepsis can lead to the apoptosis of immune cells, further depleting the body’s ability to respond to infections. This immune paralysis is a major contributor to the high mortality rate observed in sepsis patients, particularly in the later stages of the disease.

### 3.2. Immunosuppression in Cancer

Similar to sepsis, cancer exploits immunosuppression as a critical mechanism to evade immune detection and destruction, enabling tumor survival and progression. Tumors create an immunosuppressive microenvironment by recruiting regulatory T cells (Tregs) and myeloid-derived suppressor cells (MDSCs), which inhibit the activity of cytotoxic T cells and natural killer (NK) cells [31].

Tregs, which normally function to maintain immune tolerance and prevent autoimmunity, are co-opted by tumors to suppress antitumor immune responses. Tregs inhibit the activity of effector T cells through the release of immunosuppressive cytokines, such as interleukin-10 (IL-10) and transforming growth factor-beta (TGF-β), and through direct cell-to-cell contact. Similarly, MDSCs, which are immature myeloid cells, release immunosuppressive cytokines and metabolites, such as arginase and reactive oxygen species (ROS), that inhibit T-cell function and promote tumor growth [32].

The shared mechanisms of immune suppression in sepsis and cancer suggest that therapies targeting these pathways could be beneficial in both conditions. Immune checkpoint inhibitors, such as anti-PD-1 and anti-CTLA-4 antibodies, have shown remarkable efficacy in enhancing antitumor immune responses and improving patient outcomes in cancer. For example, pembrolizumab (anti-PD-1) and ipilimumab (anti-CTLA-4) have been approved for the treatment of various cancers, including melanoma and non-small-cell lung cancer [33,34].

In sepsis, immune checkpoint inhibitors are being explored as potential treatments to reverse immune paralysis and restore immune function. Preclinical studies have shown that blocking PD-1 or CTLA-4 can improve survival in animal models of sepsis by enhancing T-cell responses and reducing susceptibility to secondary infections [35]. However, the complex interplay between inflammation and immune suppression in sepsis poses significant challenges, and further research is needed to fully understand the potential benefits and risks of these therapies.

## 4. Metabolic Alterations

### 4.1. The Warburg Effect on Cancer

One of the most well-known metabolic alterations in cancer is the “Warburg effect”, named after Otto Warburg, who first observed that cancer cells preferentially rely on glycolysis for energy production, even in the presence of sufficient oxygen (aerobic glycolysis) [36]. This phenomenon is counterintuitive because glycolysis is far less efficient than oxidative phosphorylation in terms of ATP production. However, the Warburg effect provides cancer cells with several advantages. First, glycolysis generates ATP rapidly, which supports the high energy demands of rapidly proliferating cancer cells. Second, the intermediates of glycolysis can be diverted into biosynthetic pathways, providing the building blocks (e.g., nucleotides, amino acids, and lipids) needed for cell growth and division. Third, the production of lactate, a byproduct of glycolysis, creates an acidic tumor microenvironment that promotes invasion, metastasis, and immune evasion [37].

The Warburg effect is driven by oncogenic signaling pathways, such as PI3K/AKT/mTOR and MYC, which upregulate glycolytic enzymes and glucose transporters (e.g., GLUT1) [38]. Additionally, mutations in tumor suppressor genes, such as TP53, can further enhance glycolytic flux by disrupting mitochondrial function and promoting the shift toward glycolysis [39]. This metabolic reprogramming not only supports tumor growth but also contributes to the resistance of cancer cells to apoptosis and chemotherapy.

### 4.2. Metabolic Shifts in Sepsis

In sepsis, immune cells, particularly macrophages and neutrophils, also undergo a metabolic shift toward glycolysis [40,41]. This shift is essential for meeting the high energy demands of these cells as they respond to infection. Glycolysis allows immune cells to rapidly generate ATP and produce biosynthetic precursors needed for the production of cytokines, chemokines, and reactive oxygen species (ROS), which are critical for pathogen clearance.

However, this metabolic reprogramming can become dysregulated, leading to detrimental effects. In the early hyperinflammatory phase of sepsis, excessive immune cell activation drives a significant upregulation of glycolysis, contributing to tissue damage and organ dysfunction. The overproduction of lactate, a byproduct of glycolysis, contributes to lactic acidosis, a common complication of sepsis that is associated with poor outcomes [42]. Additionally, the persistent activation of glycolysis can deplete cellular energy reserves and impair mitochondrial function, leading to a state of cellular exhaustion and contributing to the later phase of immune paralysis.

### 4.3. Shared Metabolic Pathways and Therapeutic Implications

The metabolic alterations in sepsis and cancer extend beyond glycolysis and involve disruptions in the metabolism of amino acids, lipids, and other nutrients. These shared pathways highlight the interconnectedness of metabolic and immune dysregulation in both conditions.

### 4.4. Glutamine Metabolism

Glutamine, the most abundant amino acid in the blood, plays a critical role in both cancer and sepsis. In cancer, glutamine is a key nutrient that fuels the growth and survival of tumor cells, as its uptake and metabolism are essential for supporting rapid cell proliferation and biosynthesis [43,44]. Glutamine is converted to glutamate by the enzyme glutaminase (primarily GLS1/GLS2), and glutamate is further metabolized to α-ketoglutarate, which enters the tricarboxylic acid (TCA) cycle to support ATP production and various biosynthetic pathways—a process known as glutaminolysis. This pathway is particularly important in cancer cells with defective mitochondria or in hypoxic tumor microenvironments that rely on alternative metabolic routes for energy production and redox balance [45]. The glutaminase II pathway, which utilizes glutamine transaminase (GTK) and ω-amidase, is especially significant in hypoxic tumor regions because it generates α-ketoglutarate via a transamination reaction that does not require oxygen or NAD^+^, thus allowing cancer cells to maintain TCA cycle anaplerosis and biosynthesis even in low-oxygen environments [46]. This pathway is upregulated by hypoxia-inducible factors, such as HIF-2α, which also enhance the expression of glutamine transporters and enzymes, further supporting metabolic adaptation and survival in hypoxic tumor microenvironments. Notably, tumors with high glutaminase II activity, such as pancreatic and prostate cancers, often exhibit increased invasiveness and resistance to glutaminase I inhibitors, highlighting the pathway’s role in both metabolic flexibility and therapy evasion [47]. In addition to these alternative metabolic routes, specific glutamine transporters such as SLC1A5 (ASCT2) and SLC7A5 (LAT1) play crucial roles in glutamine uptake and metabolic reprogramming in both cancer and sepsis [48]. SLC1A5 mediates sodium-dependent glutamine influx, which not only fuels mitochondrial metabolism but also provides intracellular glutamine for exchange via SLC7A5, facilitating the import of essential amino acids like leucine and activating mTORC1 signaling—a major driver of tumor growth and proliferation [49]. Recent research has identified a mitochondrial variant of SLC1A5 that is critical for transporting glutamine into mitochondria, especially under hypoxic conditions, further supporting metabolic flexibility and adaptation in cancer cells. The upregulation of these transporters is often driven by oncogenes such as MYC and KRAS, and their expression is associated with aggressive tumor phenotypes and poor patient prognosis [48,50].

In sepsis, glutamine metabolism is also profoundly altered, but with distinct tissue-specific patterns [51]. Plasma concentrations of glutamine and other amino acids are typically decreased in sepsis, reflecting increased consumption and altered metabolism. Skeletal muscle, the primary glutamine reservoir, increases glutamine release during sepsis, yet the intracellular pool becomes depleted as the rate of release outpaces synthesis [51]. The liver becomes a major site of glutamine uptake, with net hepatic glutamine extraction increasing dramatically to support energy production and antioxidant synthesis—particularly glutathione, which is critical for combating oxidative stress in sepsis. Meanwhile, skeletal muscle and the spleen show suppressed glutamine contribution to the TCA cycle, and the gut exhibits decreased utilization, which may contribute to impaired citrulline production and gut barrier dysfunction. Immune cells, such as lymphocytes and macrophages, are also major glutamine consumers during sepsis, relying on glutamine for proliferation and function; glutamine availability can become rate-limiting for effective immune responses [52]. These findings underscore that, in sepsis, glutamine metabolism is dynamically reprogrammed to prioritize hepatic energy demands and antioxidant synthesis, while other tissues may experience a relative deficit, highlighting the complexity and therapeutic relevance of targeting glutamine pathways in both cancer and sepsis [52].

### 4.5. Lipid Metabolism

While glutamine metabolism fuels biosynthetic and energy demands, its interplay with lipid metabolism is equally critical for cellular survival and proliferation. Glutamine-derived α-ketoglutarate replenishes the TCA cycle, generating citrate—a precursor for acetyl-CoA, the essential building block for fatty acid synthesis. This crosstalk is amplified in cancer and sepsis, where dysregulated lipid metabolism supports membrane synthesis, signaling molecules, and energy storage. Lipid metabolism is another shared pathway that is dysregulated in both sepsis and cancer. In cancer, alterations in lipid metabolism support tumor growth, survival, and metastasis. Cancer cells often exhibit increased lipogenesis (the synthesis of fatty acids) and lipid uptake, which provide the lipids needed for membrane synthesis, signaling molecules, and energy storage [53]. Additionally, lipid metabolism plays a role in immune evasion, as lipid droplets in cancer cells can sequester pro-inflammatory signaling molecules and inhibit antitumor immune responses [54].

In sepsis, lipid metabolism is also disrupted, contributing to inflammation and immune suppression. During sepsis, there is an increase in lipolysis (the breakdown of fats), which releases free fatty acids (FFAs) into the bloodstream [55]. While FFAs can serve as an energy source, their excessive accumulation can lead to lipotoxicity, mitochondrial dysfunction, and the production of pro-inflammatory lipid mediators, such as prostaglandins and leukotrienes. These lipid mediators can exacerbate inflammation and contribute to organ damage. The shared metabolic pathways in sepsis and cancer present opportunities for targeted therapies (Table 2). For example, inhibiting glycolysis or glutaminolysis could potentially slow tumor growth in cancer and reduce the hyperinflammatory response in sepsis. Drugs that target glycolytic enzymes, such as hexokinase 2 (HK2) or lactate dehydrogenase A (LDHA), are being explored as cancer therapies and could also be tested in sepsis [56]. Similarly, inhibitors of glutaminase, such as CB-839, are being investigated in clinical trials for cancer and may have potential applications in sepsis [57]. Targeting lipid metabolism is another promising strategy. Inhibitors of fatty acid synthesis, such as FASN inhibitors, or drugs that modulate lipid signaling pathways, such as cyclooxygenase-2 (COX-2) inhibitors, could be used to disrupt the metabolic adaptations that support tumor growth and inflammation [58]. However, the complexity of metabolic networks and their dual roles in both normal and pathological processes pose significant challenges. For example, inhibiting glycolysis or glutaminolysis could impair the function of immune cells, which also rely on these pathways to fight infections.

## 5. Clinical Implications of the Sepsis–Cancer Connection

### 5.1. Impact of Sepsis on Tumor Microenvironment

The tumor microenvironment (TME) is a dynamic network of cancer cells, immune cells, stromal cells, blood vessels, and extracellular matrix (ECM) that influences tumor behavior [59]. Sepsis often leads to systemic hypoxia, a condition characterized by insufficient oxygen supply to tissues, due to microcirculatory dysfunction, impaired oxygen delivery, and mitochondrial dysfunction. Within the tumor microenvironment (TME), hypoxia activates hypoxia-inducible factor 1-alpha (HIF-1α), a master transcriptional regulator of cellular responses to low oxygen levels. HIF-1α upregulates a wide array of genes involved in angiogenesis, metabolism, and survival, enabling cancer cells to adapt to and thrive in low-oxygen conditions [60]. This adaptive response not only supports tumor survival but also contributes to its aggressiveness and resistance to therapy.

A central role of HIF-1α is the promotion of vascular endothelial growth factor (VEGF), a critical mediator of angiogenesis—the process of forming new blood vessels. VEGF stimulates the proliferation and migration of endothelial cells, leading to the formation of new blood vessels that supply tumors with oxygen and nutrients [61]. However, these newly formed vessels are often structurally abnormal, leaky, and dysfunctional, which further exacerbates hypoxia and creates a pro-tumorigenic environment. For example, in breast cancer models, sepsis-induced hypoxia has been shown to significantly increase VEGF expression, leading to enhanced tumor vascularization and metastasis [62]. This abnormal vasculature not only supports tumor growth but also facilitates the spread of cancer cells to distant organs.

Hypoxia does more than just support tumor growth; it also promotes metastasis by selecting for more aggressive cancer cell populations. Hypoxic conditions induce the expression of genes involved in invasion and migration, such as LOX (lysyl oxidase) and CXCR4 (C-X-C chemokine receptor type 4). LOX, for instance, modifies the extracellular matrix to facilitate cancer cell invasion, while CXCR4 promotes the migration of cancer cells to distant sites. In colorectal cancer, hypoxia-driven VEGF expression has been linked to increased liver metastasis, underscoring the role of sepsis-induced hypoxia in promoting tumor spread [23]. These findings highlight how hypoxia, driven by sepsis, creates a vicious cycle that fuels tumor progression and metastasis.

Moreover, the interplay between hypoxia and angiogenesis is further complicated by the immunosuppressive effects of the hypoxic TME. Hypoxia recruits immunosuppressive cells, such as myeloid-derived suppressor cells (MDSCs) and regulatory T cells (Tregs), which inhibit antitumor immune responses [63]. This immunosuppressive milieu not only allows cancer cells to evade immune detection but also creates a favorable environment for tumor growth and metastasis.

Thus, sepsis-induced hypoxia activates HIF-1α, which drives VEGF-mediated angiogenesis and promotes the expression of genes involved in invasion and metastasis [64]. While these adaptations enable cancer cells to survive in low-oxygen conditions, they also contribute to the structural and functional abnormalities of tumor vasculature, exacerbating hypoxia and creating a pro-tumorigenic environment. The resulting increase in tumor aggressiveness and metastatic potential underscores the critical role of hypoxia and angiogenesis in the sepsis–cancer connection. Understanding these mechanisms provides valuable insights into potential therapeutic targets, such as HIF-1α and VEGF inhibitors, which could disrupt this cycle and improve outcomes for patients with sepsis and cancer.

### 5.2. Inflammatory Cytokines and Reactive Oxygen Species (ROS)

Sepsis also triggers a massive release of pro-inflammatory cytokines, such as interleukin-6 (IL-6), tumor necrosis factor-alpha (TNF-α), and interleukin-1 beta (IL-1β), as well as reactive oxygen species (ROS). These molecules play a dual role in the tumor microenvironment (TME), simultaneously promoting tumor progression and suppressing antitumor immunity, thereby creating a favorable environment for cancer growth and metastasis. ROS, including superoxide anions and hydrogen peroxide, are highly reactive molecules that can directly damage cellular DNA, leading to double-strand breaks, base modifications, and chromosomal aberrations. In cancer cells, such genomic alterations can result in the activation of oncogenes or the inactivation of tumor suppressor genes, driving tumor aggressiveness and resistance to therapy. For example, in colorectal cancer, sepsis-induced ROS has been shown to promote mutations in the APC gene, a key regulator of the Wnt signaling pathway, enhancing tumor progression by disrupting normal cellular processes and promoting uncontrolled cell proliferation [65]. The accumulation of DNA damage and genomic instability not only accelerates tumor growth but also increases the likelihood of metastasis and therapy resistance, making it a critical factor in the sepsis–cancer connection.

The inflammatory milieu created by sepsis also profoundly suppresses antitumor immune responses, further facilitating cancer progression. Sepsis recruits immunosuppressive cells, such as myeloid-derived suppressor cells (MDSCs) and regulatory T cells (Tregs), to the TME. These cells play a pivotal role in creating an immune-tolerant environment by inhibiting the activity of cytotoxic T cells and natural killer (NK) cells, which are essential for detecting and destroying cancer cells. For instance, sepsis-induced IL-6 has been shown to promote the expansion of MDSCs, a heterogeneous population of immature myeloid cells that suppress T cell function and promote tumor growth. In melanoma, the expansion of MDSCs driven by sepsis-induced IL-6 has been linked to accelerated tumor growth and reduced patient survival [66]. Similarly, Tregs, which normally function to maintain immune tolerance, are co-opted by tumors to suppress antitumor immune responses. The recruitment and activation of these immunosuppressive cells create a microenvironment that allows cancer cells to evade immune surveillance and proliferate unchecked.

Inflammatory cytokines and reactive oxygen species (ROS) play a complex, dual role in the tumor microenvironment (TME), simultaneously promoting cancer progression and suppressing antitumor immunity. These mediators fuel tumor growth by inducing DNA damage, genomic instability, and activating oncogenic pathways such as NF-κB and STAT3. Concurrently, they impair host defenses by recruiting immunosuppressive cells (e.g., Tregs, MDSCs) and inhibiting effector immune cell function through mechanisms like PD-L1 upregulation. This interplay creates a self-reinforcing cycle that accelerates tumor progression and metastasis, making it a critical target for therapeutic intervention. Emerging strategies that address sepsis-associated inflammatory mediators offer promising avenues for disrupting this cycle. For instance, IL-6 signaling inhibitors like tocilizumab may reduce STAT3-driven immunosuppression, while targeted antioxidants could mitigate oxidative DNA damage without compromising essential immune signaling. These approaches underscore the potential for translating insights from sepsis biology into novel cancer immunotherapies, highlighting the interconnected nature of these seemingly distinct pathologies [67]. Additionally, therapies aimed at depleting MDSCs or blocking their immunosuppressive activity could restore antitumor immunity and improve outcomes for cancer patients with a history of sepsis.

## 6. Long-Term Consequences of Sepsis in Cancer Patients

Sepsis has profound and lasting effects on cancer patients, creating a conducive environment for cancer progression and recurrence through mechanisms such as chronic inflammation, immune suppression, genomic instability, and epigenetic modifications. Persistent inflammation post-sepsis, characterized by the sustained release of pro-tumorigenic cytokines like IL-6 and TNF-α, fuels tumor growth and immune evasion, increasing the risk of recurrence. For example, in ovarian cancer patients, post-sepsis chronic inflammation has been correlated with higher rates of tumor recurrence, underscoring the role of inflammation in driving cancer progression [68]. Sepsis-induced immune paralysis further exacerbates this issue by compromising antitumor immunity. The dysfunction of key immune cells, such as T cells, natural killer (NK) cells, and dendritic cells, impairs the body’s ability to detect and eliminate cancer cells. In melanoma patients, sepsis-induced immune suppression has been associated with accelerated tumor progression and reduced survival, highlighting the detrimental impact of immune dysfunction on cancer outcomes [29]. Additionally, sepsis can directly damage cellular DNA through the overproduction of reactive oxygen and nitrogen species (ROS/RNS), leading to double-strand breaks, base modifications, and other forms of DNA damage. In cancer cells, these alterations result in genomic instability and the accumulation of oncogenic mutations, such as those in the APC gene in colorectal cancer, which drive tumor aggressiveness and therapy resistance [65]. Sepsis also induces epigenetic changes, including DNA methylation and histone acetylation, which alter gene expression patterns without modifying the DNA sequence itself. For instance, sepsis-induced hypermethylation of the PTEN gene, a tumor suppressor, has been linked to tumor growth and metastasis in prostate cancer [69]. Together, these long-term consequences of sepsis—chronic inflammation, immune suppression, genomic instability, and epigenetic modifications—create a pro-tumorigenic environment that not only promotes cancer progression but also increases the risk of recurrence, emphasizing the need for targeted therapeutic strategies to mitigate these effects and improve outcomes for cancer patients with a history of sepsis. Several confounding factors significantly influence sepsis-related cancer outcomes and complicate patient management. Chemotherapy and radiation, both mainstays of cancer therapy, exacerbate sepsis-induced immunosuppression by impairing mucosal barriers, reducing neutrophil counts, and altering lymphocyte function. Chemotherapy-induced neutropenia, for example, increases susceptibility to sepsis, while radiation-induced epithelial damage facilitates bacterial translocation, compounding immune paralysis [70]. Immunosuppressants, including corticosteroids and immune checkpoint inhibitors such as PD-1 blockers, further modulate the interplay between sepsis and cancer in complex ways. While PD-1 inhibition can reverse T cell exhaustion in sepsis, it may paradoxically amplify immunosuppressive regulatory T cells (Tregs) and myeloid-derived suppressor cells (MDSCs) in comorbid patients, potentially leading to shorter progression-free survival when used concurrently with immunotherapy [71]. The stage of cancer and the patient’s baseline immune status also play crucial roles; advanced cancer stages are associated with profound immunosuppression, including elevated Tregs and altered neutrophil-to-lymphocyte ratios, which, when combined with sepsis-driven myeloid dysfunction, can accelerate tumor growth. Post-septic states in these patients often show reduced tumor-infiltrating CD8+ T cells and diminished MHC-I expression, enabling further immune evasion [72]. Additional comorbidities, such as diabetes or COPD, can amplify both sepsis severity and cancer progression by promoting chronic inflammation and compromising epithelial barriers, while older patients face higher mortality due to diminished immune resilience and preexisting organ dysfunction [73]. The timing of sepsis relative to cancer diagnosis also matters: preclinical models demonstrate that sepsis occurring before cancer can accelerate tumor growth through sustained immunosuppression, whereas sepsis after cancer may transiently inhibit tumors via NK cell activation but later drive recurrence through IL-10/PD-L1-mediated immune exhaustion [72]. Collectively, these factors underscore the complexity of managing sepsis in cancer patients and highlight the necessity for individualized therapeutic strategies that account for treatment history, immune status, comorbidities, and disease stage.

## 7. Therapeutic Opportunities

The shared mechanisms between sepsis and cancer present unique opportunities for therapeutic intervention, with several promising strategies emerging from the overlapping pathways of inflammation, immune dysregulation, and metabolic reprogramming. Anti-inflammatory therapies, such as targeting cytokines like IL-6 and TNF-α, have shown potential in mitigating inflammation in both conditions. For instance, monoclonal antibodies like tocilizumab (anti-IL-6) have been effective in dampening the cytokine storm in sepsis and are being explored in cancer, while TNF-α inhibitors like infliximab are used in autoimmune diseases and are being investigated for their ability to reduce inflammation-driven tumor progression. However, the dual roles of these cytokines in both protective and pathological processes pose challenges, as their inhibition may compromise the body’s ability to fight infections or control tumor growth. Immune checkpoint inhibitors, such as anti-PD-1 (e.g., pembrolizumab) and anti-CTLA-4 (e.g., ipilimumab) antibodies, have revolutionized cancer treatment by reactivating antitumor immunity and are now being explored in sepsis to reverse immune paralysis and restore immune function. Preclinical studies have demonstrated that blocking PD-1 or CTLA-4 can improve survival in sepsis models by enhancing T-cell responses, though overactivation of the immune system remains a concern. Metabolic modulators, including inhibitors of glycolysis (e.g., 2-deoxy-D-glucose) and glutaminolysis (e.g., CB-839), offer another therapeutic avenue. These drugs target the metabolic reprogramming seen in both cancer cells, which rely on aerobic glycolysis (the Warburg effect), and immune cells in sepsis, which undergo a glycolytic shift to meet energy demands during infection. However, the dual roles of these pathways in normal and pathological processes necessitate careful consideration, as their inhibition could impair immune cell function. In Table 3, we list the drugs that have been studied in both sepsis and cancer models for research. Antimicrobial peptides (AMPs), such as cathelicidins and defensins, represent a versatile therapeutic option due to their antimicrobial and antitumor properties [74,75,76]. In sepsis, AMPs such as LL-37 can effectively neutralize pathogens by disrupting microbial membranes and modulating the immune response to reduce inflammation [77]. In cancer, AMPs exhibit selective cytotoxicity toward cancer cells, inhibiting key processes like angiogenesis and tumor growth. Their ability to distinguish between malignant and normal cells makes them valuable candidates for targeted cancer therapy. Several peptides derived from the naturally occurring proteins, including Myeloid Differentiation Factor 2 (MD2) [78,79], MyD88 [80], MIEN1 [81,82], and Annexin A2 [83,84], have been identified to possess either anticancer or anti-inflammatory properties—or both. These peptides can be leveraged to develop dual therapeutics that target the shared inflammatory and immune pathways involved in both sepsis and cancer. For example, MD2 and MyD88 play key roles in the Toll-like receptor (TLR) signaling pathway, which regulates immune responses to pathogens and tumors. MIEN1 and Annexin A2 have been implicated in cancer progression and metastasis, but their modulation can also dampen inflammatory responses, suggesting potential for integrated sepsis and cancer therapies [85,86]. Structural motifs within proteins, such as phenylalanine zippers and conserved motifs like GXXXXG, offer valuable tools for designing synthetic peptides with enhanced specificity and stability [87,88]. Phenylalanine zippers can stabilize peptide structures, increasing their ability to penetrate cell membranes and improve therapeutic efficacy [89]. Similarly, the GXXXXG motif has been shown to enhance the antimicrobial and anticancer activity of certain peptides while reducing cytotoxicity toward normal cells. These structural elements can be engineered as molecular “switches” to create dual-function peptides that simultaneously exhibit anti-endotoxin and anti-inflammatory properties [90]. Moreover, AMPs enhance antitumor immunity by recruiting and activating immune cells such as macrophages, natural killer (NK) cells, and T cells. This immunomodulatory effect increases the body’s ability to recognize and eliminate malignant cells, further enhancing the therapeutic potential of AMPs in cancer treatment. However, despite their promise, challenges remain in optimizing the stability, specificity, and delivery of AMPs for clinical applications. Issues such as peptide degradation in physiological conditions, off-target effects, and limited bioavailability need to be carefully addressed to translate preclinical successes into clinical outcomes.

Together, these strategies underscore the potential for integrated approaches to target the shared pathophysiology of sepsis and cancer. By leveraging the dual roles of AMPs and naturally occurring proteins, it may be possible to develop novel, multi-functional therapies that not only treat infections and inflammation but also inhibit tumor progression. However, a careful balance of these mechanisms is essential to maximize therapeutic benefits while minimizing potential risks.

## 8. Conclusions

The emerging evidences clearly reveal that sepsis and cancer, though distinct in their clinical presentations, are deeply interconnected through shared biological mechanisms, including inflammation, immune dysregulation, and metabolic reprogramming. These overlapping pathways not only drive disease progression but also present unique opportunities for therapeutic intervention. The shared mechanisms between these conditions offer promising avenues for targeted therapies. Anti-inflammatory agents, such as IL-6 and TNF-α inhibitors, have shown potential in mitigating inflammation in both sepsis and cancer, though their dual roles in protective and pathological processes require careful consideration. Immune checkpoint inhibitors, including anti-PD-1 and anti-CTLA-4 antibodies, have revolutionized cancer treatment and are now being explored in sepsis to reverse immune paralysis and restore immune function. Metabolic modulators, such as glycolysis and glutaminolysis inhibitors, target the metabolic reprogramming common to both conditions, though their impact on normal cellular functions must be carefully balanced. Antimicrobial peptides (AMPs), with their dual antimicrobial and antitumor properties, represent a versatile therapeutic option, though challenges in stability and delivery remain. Understanding the intricate interplay between sepsis and cancer is critical for developing integrated therapeutic strategies that address their shared pathophysiology. By targeting overlapping mechanisms, such as inflammation, immune suppression, and metabolic alterations, we can improve outcomes for patients affected by these devastating diseases. However, the dual roles of these pathways in both protective and pathological processes highlight the need for a nuanced approach to therapy, one that balances the need to control disease progression without compromising essential immune and metabolic functions. Future research should focus on elucidating the context-specific roles of these shared mechanisms and developing personalized therapies that account for the complex interplay between sepsis and cancer. By bridging the gap between sepsis and cancer research, we can inspire interdisciplinary collaboration and advance the development of innovative diagnostic tools and therapeutic strategies. Ultimately, this integrated approach has the potential to transform the management of sepsis and cancer, improving survival and quality of life for patients worldwide.

## Figures and Tables

**Figure 1 biomedicines-13-01280-f001:**
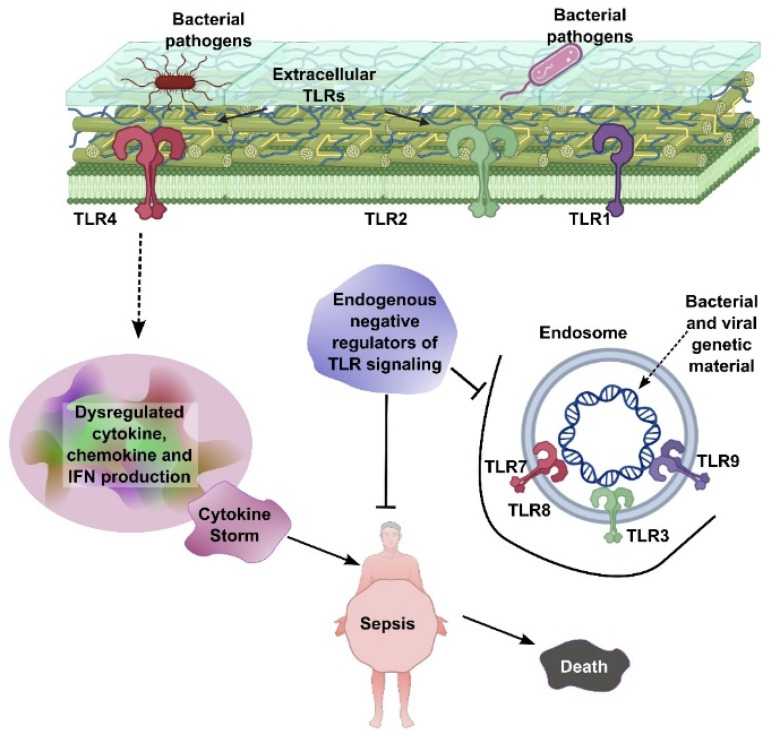
TLR-mediated sepsis pathogenesis. Extracellular TLRs (TLR1, TLR2, and TLR4) recognize bacterial pathogens, leading to dysregulated cytokine, chemokine, and IFN production and subsequent cytokine storm. Endosomal TLRs (TLR3, TLR7, TLR8, and TLR9) detect bacterial/viral genetic material, further contributing to inflammation. Endogenous negative regulators of TLR signaling aim to control this process, but failure leads to sepsis and death.

**Figure 2 biomedicines-13-01280-f002:**
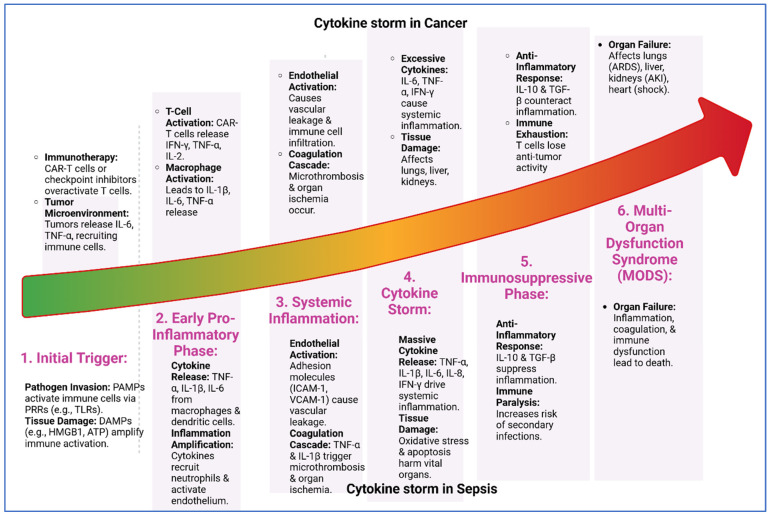
Cytokine storm progression in cancer and sepsis. The six shared stages of cytokine storm in cancer and sepsis: (1) initial trigger by pathogens or tumor signals, (2) early pro-inflammatory phase with cytokine release, (3) systemic inflammation causing vascular leakage and microthrombosis, (4) cytokine storm with excessive cytokines and tissue damage, (5) immunosuppressive phase marked by anti-inflammatory responses and immune exhaustion, and (6) multi-organ dysfunction syndrome (MODS) leading to organ failure and death. Despite different triggers, both conditions follow a similar inflammatory trajectory.

**Table 1 biomedicines-13-01280-t001:** Cytokines linking sepsis and cancer: common cytokines involved in both conditions, with their roles in inflammation, immune regulation, and disease progression.

Cytokine	Role in Sepsis	Role in Cancer
TNF-α	-Early pro-inflammatory mediator. -Promotes inflammation, endothelial activation, and organ dysfunction.	-Promotes tumor progression, angiogenesis, and metastasis. -Can induce tumor cell death in high concentrations.
IL-1β	-Induces fever, vasodilation, and immune cell recruitment. -Contributes to tissue damage.	-Promotes tumor growth, angiogenesis, and metastasis. -Enhances immunosuppressive microenvironment.
IL-6	-Key mediator of acute-phase response. -Correlates with severity and mortality.	-Promotes tumor growth, survival, and metastasis. -Drives chronic inflammation and immune evasion.
IL-8 (CXCL8)	-Chemoattractant for neutrophils. -Contributes to tissue injury.	-Promotes angiogenesis, tumor growth, and metastasis. -Attracts immunosuppressive cells to the tumor microenvironment.
IL-10	-Anti-inflammatory cytokine. -Suppresses pro-inflammatory responses. -Can lead to immunosuppression.	-Promotes immune evasion by suppressing antitumor immunity. -Enhances tumor progression.
IL-17	-Produced by Th17 cells. -Promotes neutrophil recruitment and inflammation.	-Promotes tumor growth, angiogenesis, and metastasis. -Contributes to chronic inflammation.
IL-23	-Promotes Th17 cell differentiation and IL-17 production. -Amplifies inflammation.	-Promotes tumor growth and immune evasion. -Enhances chronic inflammation.
IFN-γ	-Activates macrophages and enhances pro-inflammatory responses. -Contributes to tissue damage.	-Can have antitumor effects by activating immune cells. -May promote tumor immune evasion in chronic settings.
TGF-β	-Anti-inflammatory cytokine. -Promotes tissue repair and immunosuppression.	-Promotes tumor progression, immune evasion, and metastasis. -Induces epithelial–mesenchymal transition (EMT).
VEGF	-Promotes vascular permeability and endothelial dysfunction.	-Drives angiogenesis, supporting tumor growth and metastasis.
HMGB1	-Late-phase mediator of sepsis. -Sustains inflammation and organ damage.	-Promotes tumor growth, metastasis, and immune evasion. -Acts as a damage-associated molecular pattern (DAMP).
PD-1/PD-L1	-Contributes to T-cell exhaustion and immunosuppression in sepsis.	-Key immune checkpoint in cancer. -Promotes immune evasion and tumor progression.
G-CSF	-Stimulates neutrophil production and mobilization.	-Promotes tumor growth and metastasis. -Enhances myeloid-derived suppressor cells (MDSCs).
MCP-1 (CCL2)	-Recruits monocytes and macrophages to sites of inflammation.	-Recruits tumor-associated macrophages (TAMs), promoting tumor progression and immune evasion.

**Table 2 biomedicines-13-01280-t002:** Metabolic similarities in sepsis and cancer. Comparison of key metabolic changes in sepsis and cancer, emphasizing shared pathways and clinical implications for potential therapeutic strategies.

Aspect	Sepsis	Cancer	Clinical Implications
**Glucose Metabolism**	-**Hyperglycemia**: Insulin resistance and increased gluconeogenesis. -**Warburg Effect**: Increased glycolysis.	-**Warburg Effect**: Aerobic glycolysis for rapid ATP production. -**Increased Glucose Uptake**: Enhanced by GLUT transporters.	-**Targeting Glycolysis**: Inhibitors like 2-DG may help in both conditions. -**Glucose Control**: Tight glucose management improves outcomes in sepsis.
**Lactate Production**	-**Lactic Acidosis**: Excessive glycolysis leads to lactate accumulation.	-**High Lactate Levels**: Lactate contributes to tumor microenvironment acidosis.	-**Lactate as a Biomarker**: High lactate levels correlate with poor prognosis in both conditions.
**Lipid Metabolism**	-**Lipolysis**: Increased breakdown of fats for energy. -**Hyperlipidemia**: Elevated free fatty acids.	-**Lipid Synthesis**: Increased de novo lipogenesis. -**Fatty Acid Oxidation**: Some cancers rely on fatty acids for energy.	-**Lipid-Targeting Therapies**: Inhibitors of lipogenesis (e.g., FASN inhibitors) are explored in cancer.
**Protein Metabolism**	-**Protein Catabolism**: Muscle breakdown for gluconeogenesis. -**Negative Nitrogen Balance**.	-**Increased Protein Synthesis**: Supports cell proliferation. -**Amino Acid Dependency**: Reliance on glutamine.	-**Nutritional Support**: Glutamine supplementation may benefit both conditions.
**Glutamine Metabolism**	-**Glutamine Utilization**: Supports immune cell function and energy production.	-**Glutamine Addiction**: Used for anaplerosis and nucleotide synthesis.	-**Glutaminase Inhibitors**: CB-839 is being tested in cancer and may have potential in sepsis.
**Mitochondrial Dysfunction**	-**Impaired Oxidative Phosphorylation**: Reduced ATP production. -**ROS Production**: Contributes to tissue damage.	-**Altered Mitochondrial Function**: Dysfunction or upregulation depending on cancer type. -**ROS Signaling**: Promotes tumor growth.	-**Antioxidant Therapies**: May help mitigate ROS-induced damage in both conditions.
**Ketone Body Metabolism**	-**Increased Ketogenesis**: In response to energy demands.	-**Ketone Utilization**: Some cancers use ketone bodies as an energy source.	-**Ketogenic Diets**: May benefit cancer patients and potentially sepsis patients.
**Immune Cell Metabolism**	-**Metabolic Reprogramming**: Immune cells shift to glycolysis. -**Immunosuppression**: M2 macrophages rely on oxidative metabolism.	-**Tumor-Associated Immune Cells**: TAMs and Tregs exhibit metabolic changes supporting tumor growth.	-**Immunometabolism Targeting**: Modulating immune cell metabolism may improve outcomes.
**Hypoxia Response**	-**HIF-1α Activation**: Promotes glycolysis and angiogenesis.	-**HIF-1α Activation**: Drives angiogenesis and tumor progression.	-**HIF-1α Inhibitors**: Potential therapeutic target in both conditions.
**Insulin Resistance**	-**Peripheral Insulin Resistance**: Reduces glucose uptake in muscle and adipose tissue.	-**Altered Insulin Signaling**: Some cancers exhibit insulin resistance or upregulate insulin/IGF-1 signaling.	-**Insulin Sensitizers**: May improve outcomes in sepsis and certain cancers.
**Acidosis**	-**Metabolic Acidosis**: Due to lactate accumulation and impaired renal function.	-**Tumor Microenvironment Acidosis**: Results from high lactate production.	-**pH Modulation**: Alkalinizing agents may help mitigate acidosis in both conditions.
**Energy Demand**	-**Increased Energy Demand**: To support hypermetabolic state and immune responses.	-**Increased Energy Demand**: To support rapid cell proliferation and tumor growth.	-**Nutritional Support**: High-calorie diets may benefit patients in both conditions.

**Table 3 biomedicines-13-01280-t003:** Drugs used in both sepsis and cancer models: listed drugs, which have shown therapeutic potential in both sepsis and cancer across various experimental and clinical models.

Therapy	Indication	Clinical Trial/Approval	Outcome/Status	Reference
Tocilizumab	Cancer (CRS)	FDA Approval (2017)	Approved for CAR-T CRS	[91]
Tocilizumab	Sepsis (children)	Retrospective study	Reduced mortality, small number of cases	[92]
Infliximab	Cancer (NSCLC)	NCT00058264	No benefit, increased fatigue	[93]
Infliximab	Sepsis	Observational	Infection risk, not recommended	[94]
Checkpoint Inhibitors	Sepsis	NCT02576457 (Phase 1b)	Safe, immunomodulatory, efficacy unproven	[95]

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
