# Peer review of "The Overlapping Biology of Sepsis and Cancer and Therapeutic Implications"

_biomedicines, 2025, doi:10.3390/biomedicines13061280_

Round 1
Reviewer 1 Report
Comments and Suggestions for Authors
Thank you for the invitation to review the publication " The Overlapping Biology of Sepsis and Cancer and Therapeutic Implications". This article deals with a significant aspect of molecular medicine, connecting biology between apparently different clinical entities. The ultimate goal of these studies is to elucidate common molecular pathways, implicating in identifying possible therapeutic targets for both entities. In this aspect, interactions in molecular pathways have been identified between cardiovascular and autoimmune diseases, endometriosis and cardiovascular diseases and even overlapping mechanisms between different autoimmune diseases (e.g Systemic Lupus Erythematosus and Rheumatoid arthritis).
The study, in concordance with the aforementioned information, summarizes the intricate biological mechanisms linking these two conditions, focusing on the roles of inflammation, immune dysregulation, and metabolic alterations.
The paper's organization is coherent, and the design of the reviewing process seems rigorous.
The topic is clinically significant and has the potential to influence understanding and therapeutic strategies in both diseases. The methodology is very well applied The manuscript is well-organized and readable, with a clear flow from background to conclusion.
The tables proficiently synthesize the detailed data, rendering the study's conclusions more comprehensible to readers.
I propose the acceptance of the paper in its current form.
Author Response
Thank you very much for your thoughtful and encouraging review of our manuscript.We truly appreciate your positive feedback.
Reviewer 2 Report
Comments and Suggestions for Authors
First, I would like to acknowledge all the efforts made in this review article, and I find it
important, structured, and well-written.
This review highlights the overlap between cancer and sepsis at different levels, such as
inflammation, immune dysregulation and metabolic reprogramming.
The authors thoroughly discuss these interconnected mechanisms and point to several therapeutic agents that could target these mechanisms in both diseases.
Although there are many studies that discuss the common features between sepsis and cancer,
particularly regarding the immune system, I can consider this article as a one-stop library for
previous research and a valuable source of information about cancer-sepsis therapy.
Author Response
Thank you so much for the thoughtful and encouraging feedback. We're really glad to hear that you found the review structured, informative, and valuable.
Reviewer 3 Report
Comments and Suggestions for Authors In the review manuscript titled "The Overlapping Biology of Sepsis and Cancer and Therapeutic Implications” by Tripathi et. al. the authors investigate the biological links between sepsis and cancer, focusing on shared pathways like inflammation, immune dysfunction, and metabolic reprogramming. They explore potential therapies targeting these overlaps, such as immune checkpoint inhibitors and metabolic modulators with the goal to promote integrated research and develop targeted treatments to improve outcomes for patients with both conditions. It's an interesting review. However, there are major concerns that need to be addressed before the manuscript could be considered for publication. 1. Epidemiology of sepsis repeated in lines 43 and 46. 2. Line 53, the authors should include references where they mention emerging evidence. 3. Lines 171-174 copied as lines as 194-196. It seems like reading two parallel reviews (sepsis and cancer) rather than an integrated analysis of their overlapping biology. There is minimal discussion on how these shared mechanisms can inform dual-purpose therapies or present risks when treating comorbid cases. 4. The review mentions several studies but often lacks context for how these studies support or contrast each other. For example, the sentence "The glutaminase II pathway, which has been recently identified in some cancer types, provides an alternative route for glutamine metabolism and may be particularly important in hypoxic tumor regions " could be expanded to explain why this pathway is significant compared to the glutaminase I pathway or other metabolic routes in hypoxic conditions. 5.The section could benefit from a more in-depth explanation on how specific enzymes like SLC1A5 or SLC7A5 contribute to glutamine uptake. While these transporters are mentioned, their mechanisms and contributions to metabolic reprogramming in cancer and sepsis could be expanded upon to give readers a better understanding of their relevance. 6.The review jumps between concepts without always providing clear transitions. For instance, after discussing the role of glutamine in immune cell function during sepsis, it shifts to lipid metabolism without clear segmentation or explanation of why the transition is made. Adding a few sentences to tie these sections together could improve the readability. 7. While the review focuses a lot on cancer metabolism, the section on sepsis could be expanded with more detail on how sepsis specifically alters glutamine metabolism beyond immune dysfunction. Sepsis-induced tissue damage and the depletion of glutamine in specific organ systems could be discussed in greater detail. 8. With respect to section 7, there’s no discussion of confounding factors (e.g., chemotherapy, immunosuppressants, cancer stage), which limits the interpretability of the associations.9. The authors are mentioning reference promising drugs (e.g., tocilizumab, infliximab, checkpoint inhibitors) but didn't mention clinical trial phases, success/failure, or translational barriers. This could mislead readers into thinking these therapies are ready for dual application.
Author Response
We sincerely thank you for taking the time to provide such thoughtful and constructive feedback. Your comments were not only insightful but also genuinely helped us improve the clarity, flow, and depth of our manuscript.
1. Comment: Epidemiology of sepsis repeated in lines 43 and 46.
Response:
We thank the reviewer for pointing this out. The repeated statements regarding the epidemiology of sepsis in lines 43 and 46 have been removed to avoid redundancy and improve clarity.
2. Comment: Line 53, the authors should include references where they mention emerging evidence.
Response:
Appropriate references have been added to support the statement regarding emerging evidence. This addition strengthens the section and provides readers with reliable sources. The new references are 3 and 4 in the revised manuscript.
3. Comment: Lines 171–174 copied as lines 194–196.
Response:
We apologize for the oversight. The duplicated content in lines 194–196 has been removed, and the section has been revised to ensure a smoother flow and avoid repetition.
4. Comment: It seems like reading two parallel reviews (sepsis and cancer) rather than an integrated analysis of their overlapping biology. There is minimal discussion on how these shared mechanisms can inform dual-purpose therapies or present risks when treating comorbid cases.
Response:
The manuscript has been restructured to present a more integrated analysis. We focused on highlighting the overlapping biological mechanisms between sepsis and cancer, particularly within lines 239–300. We also added new content discussing how these shared pathways may lead to dual-purpose therapies and what risks might arise when treating patients with both conditions.
5. Comment: The review mentions several studies but often lacks context for how these studies support or contrast each other. For example, the sentence about the glutaminase II pathway could be expanded.
Response:
Thank you for raising this excellent point We have expanded the discussion on the glutaminase II pathway to explain how it compares to the glutaminase I pathway, especially under hypoxic conditions. Additional context has also been provided throughout the manuscript to show how various studies align or differ, offering a clearer narrative.
6. Comment: The section could benefit from a more in-depth explanation on how specific enzymes like SLC1A5 or SLC7A5 contribute to glutamine uptake.
Response:
A very valid point from the reviewer. Further explanation has been added regarding the roles of SLC1A5 and SLC7A5 in glutamine transport. Their mechanisms and contributions to metabolic reprogramming in cancer and sepsis have been detailed to give readers a better understanding of their relevance.
7. Comment: The review jumps between concepts without always providing clear transitions, e.g., from glutamine in immune cells to lipid metabolism.
Response:
We are very sorry for not being careful enough on this. Transitions between major topics have been improved to create smoother section flow. Linking sentences were added, particularly between discussions on glutamine metabolism and lipid metabolism in Sections 4.4 and 4.5, to help readers follow the shift in concepts more easily.
8. Comment: While the review focuses a lot on cancer metabolism, the section on sepsis could be expanded with more detail on how sepsis specifically alters glutamine metabolism.
Response:
We expanded the discussion of glutamine metabolism in sepsis to include tissue-specific depletion, impacts on organ systems, and the role of glutamine in recovery from tissue damage. These additions help to better illustrate how glutamine metabolism is disrupted in sepsis beyond its effects on immune cells. Please see lines 276-293.
10. Comment: Promising drugs are mentioned (e.g., tocilizumab, infliximab, checkpoint inhibitors) but their clinical status isn't clarified, which could mislead readers.
We appreciate this observation and have clarified the clinical relevance of the mentioned therapies by adding a new summary table (Table 3) that lists each drug’s clinical trial phase, approval status, and reported outcomes in both sepsis and cancer, along with clinical trial identifiers and key references. Additionally, a new section has been included to outline the translational challenges of using these therapies in both conditions, highlighting differences in mechanisms, safety concerns, and the absence of predictive biomarkers. To avoid any misunderstanding, we have also clearly stated that while these agents are promising, their dual application is still investigational and not yet established for clinical use.
Round 2
Reviewer 3 Report
Comments and Suggestions for Authors
The authors have addressed all my concerns and have made suggested changes in the manuscript. I, therefore, recommend the manuscript suitable for publication in its revised form.